# Research on the Applicability of Vibration Signals for Real-Time Train and Track Condition Monitoring [note 1]

**DOI:** 10.3390/s22062368

**Published:** 2022-03-18

**Authors:** Ireneusz Celiński, Rafał Burdzik, Jakub Młyńczak, Maciej Kłaczyński

**Affiliations:** 1Department of Transport Systems, Traffic Engineering and Logistics, Faculty of Transport and Aviation Engineering, Silesian University of Technology, 40-019 Katowice, Poland; ireneusz.celinski@polsl.pl; 2Department of Road Transport, Faculty of Transport and Aviation Engineering, Silesian University of Technology, 40-019 Katowice, Poland; 3Department of Rail Transport, Faculty of Transport and Aviation Engineering, Silesian University of Technology, 40-019 Katowice, Poland; jakub.mlynczak@polsl.pl; 4Department of Mechanics and Vibroacoustics, Faculty of Mechanical Engineering and Robotics, AGH University of Science and Technology, 30-059 Cracow, Poland; maciej.klaczynski@agh.edu.pl

**Keywords:** statistical analysis, train vibration, resultant vector, track in service diagnostics

## Abstract

The purpose of this research was to analyze the possibilities for the application of vibration signals in real-time train and track control. Proper experiments must be performed for the validation of the methods. Research on vibration in the context of transport must entail many of the different nonlinear dynamic forces that may occur while driving. Therefore, the paper addresses two research cases. The developed application contains the identification of movement and dynamics and the evaluation of the technical state of the rail track. The statistics and resultant vector methods are presented. The paper presents other useful metrics to describe the dynamical properties of the driving train. The angle of the resultant horizontal and vertical accelerations is defined for the evaluation of the current position of cabin. It is calculated as an inverse tangent function of current longitudinal and transverse, longitudinal and vertical, transverse, and vertical accelerations. Additionally, the resultant vectors of accelerations are calculated.

## 1. Introduction

The current development, applications, and implications in the field of transportation are the main focus of transport scientists and engineers. The possibilities and impacts of emerging technologies, such as vibro-acoustics, on transportation system performance, in terms of levels of safety and reliability, are widely discussed in many papers. Vibration phenomena in transport engineering can be considered as unwanted effects, mostly in terms of human exposure and as the source of degradation of the infrastructure and suprastructure of a transport system. However, it should be considered as a source of much information on the technical condition of transport methods and the transport network. Vibrations, very often of a non-linear nature, are generated during the operation of machines and devices and transferred via hands or other parts of the body; these include the feet and the lumbar region, when in a sitting position. Long-term exposure to vibrations may adversely affect human health [1,2,3,4]. The negative effects (pathologies) of long-term exposure to mechanical vibrations may be permanent or even irreversible. To a large extent, this risk depends on the frequency of vibrations, the exposure time, and the contact surface in the human–device system [3,4,5,6]. Conditions related to the negative influence of acceleration are generally called kinetosis [7]. The term “vibration stress” is also often encountered. In this way, the human body responds naturally to selected movement stimulants [4]. On the other hand, vibrations transfer much information. A wide range of frequency capacities and non-destructive methods of vibration measurement are commonly used in machine monitoring systems [8,9,10,11,12,13,14,15,16,17,18]. 

Paper [19] describes research on vibrations in the cab of the train driver during shunting activity. The authors extended the research and analysis to explore the applicability of the vibration signals of the train driver’s cab for train and track control. For this reason, in this article, the authors have addressed and discussed selected aspects of the analysis of the vibration impact on the driver of a shunting locomotive and an EN57 suburban multiple-unit set.

One group of applications of vibration research refers to the evaluation of human exposure to vibration, which was the main focus of [1,2,3,4,5]. In publications concerning this subject, problems arising from the impact of vibrations on people are mainly analyzed with regard to manually operated machines and devices, due to the scale of the phenomenon. In cases wherein the vibration impact affects the entire body, it may primarily cause dysfunction of the organs of the abdominal cavity (especially the stomach), and it also affects the abdomen in women. Other effects observed in practice on people operating vibrating systems include often prolonged fatigue, headaches, sleep dysfunction, and trembling limbs; these symptoms are typical of those operating vibrating equipment. However, in the case discussed, that is, the whole-body impact of vibrations, this problem may not only concern the hands and arms [1,4]. Prolonged exposure of the human body to vibrations may trigger the spread of disorders to other internal organs. It has been confirmed that vibrations can cause diseases of the cardiovascular, respiratory, digestive, and musculoskeletal systems [6,7]. Vibrations that affect the entire body of a machine operator may also lead to changes in pulse, as well as in the vascular and urinary systems. In addition to the direct hazard to the health and life of a worker exposed to whole-body vibrations, one can also observe a reduction in immunity. Therefore, from the employer’s perspective, which involves maintaining continuity and operating safety, hazards connected with vibrations occurring in railway vehicles (or, more broadly, in transport) can be divided into: short-term, which can involve hazards to traffic safety in certain circumstances, and long-term, causing a deterioration of operating efficiency and loss of highly qualified workers for prolonged periods due to sick leave [6].

Probably the largest number of publications on vibration tests in the field of technical sciences refer to applications in diagnostics and monitoring of technical condition. Applications of vibration methods in monitoring the condition of motor vehicle systems are more and more often presented [8,9,10,11]. Many publications focus on new advanced signal processing methods in order to recognize the condition of a technical object [12,13,14,15,16,17,18]. The paper [12] presents an application of generalised entropy and fractal diagnostic models of mechanical vibration signals. The statistical modeling of diagnostic data has been introduced in [13]. Very often, in signal processing, multivariate analyzes, including time-frequency transformations, are used [14,15,16]. This enables the observation of time-varying nonstationary signal components of nonlinear systems [17,18]. An interesting approach by applying TRIZ methodology for vibration condition has been presented in [19].

Vibration research in rail transport is most often related to the phenomenon of wheel-rail contact or impact on buildings in the vicinity of the tracks [20,21,22,23]. On the other hand, a very frequently studied problem in rail transport is the control and management of vehicle traffic [24,25,26,27]. Thus, vibroacoustic methods should be considered as a support for monitoring systems in the field of transportation due to their usefulness in evaluating human exposure and monitoring the technical state of the transport system. The article presents implications of the methodologies developed for the rail transport system for operation, control, and maintenance services.

Vibrations are also used in systems for monitoring and diagnosing the condition of tracks and the bogies of rail vehicles. The vibration of a train track under a train in motion provides important information about the train and the track. Traditionally, track monitoring systems are used for wheel-impact load detection, continuous monitoring of the wheel profile, bogie performance, etc. [28]. Paper [29] presents the concept of a simple sensor-based system for train condition monitoring. This system uses an ADXL335; this is a small, thin, low-power, complete 3-axis accelerometer with signal-conditioned voltage output. The fundamental time-domain and frequency-domain based methods have been used to determine the condition of a train using measurements of railway track vibrations under a train in motion. The article in [29] confirms the legitimacy of searching for new, low-cost solutions that can be used in the diagnosis of tracks and rail vehicles. The authors conducted many studies using rail vibration measurements with a view to identifying passing trains [30] and the vibration responses of rail infrastructure devices or passing cars [31], as well as identifying rail damage [32].

Another approach is to record vibrations in moving locomotives or wagons. Paper [33] presents the current state of the art, and reviews recent on-board condition monitoring sensors and techniques and potential applications for integrating condition-monitoring systems into freight wagons. Paper [34] presents a simple and effective fault-detection framework for the polygonal wear of the railway wheel. It is based on easily acquired axle-box vibration acceleration data to detect both the level and the order parameters of the polygon wear of railway wheel. Paper [35] presents numerical experiments on the method, which provides a useful reference source for the detection of polygonization for high-speed trains, as well as a judgement of the validity of dynamic response data. Another approach to vibration-based damage identification and condition monitoring of metro trains by the system installed in buildings has been presented in [36]. The vibration is triggered by a signal transmitted from the measuring point located in the tunnel, which triggers the registration with a photocell when passing a subway train. Therefore, all the vibration methods used in the research on rail vehicles and tracks are based on dynamic dependencies and vibration signals generated as a result of them [37,38].

When considering the use of vibration signals for diagnostic purposes, acoustic measurements should also be considered as an additional source of information [39]. In this case, vibroacoustic methods can be applied. Paper [40] presents the application of distributed acoustic sensor technology to monitor the running condition of trains. The new method of monitoring train running conditions using distributed acoustic sensor (DAS) technology is based on the fact that the vibration of the environment around the optical fiber will change when the train passes [41]. This method is very effective, but requires a lot of money and cannot always be used. Therefore, the authors decided to carry out measurement experiments under the assumption of minimal measurement and equipment costs.

Railway traffic is enormously influenced by the variation in contact stress caused by wheel–rail contact profile changes [42]. Paper [43] presents a study on the influence of the topology of standard rail and wheel profiles. The condition of the wheel sets also significantly influences the vibrations generated by the rail vehicle. Paper [44] especially considers abrasive wear of the wheel rims and damage of the spalling type on the wheel thread in wheel sets. Therefore, the importance of the phenomenon of wheel rail contact—which is the subject of many scientific publications—should be emphasized in research on rail vehicle vibrations.

Other types of vibration research focus on the selection of useful information about the current condition of the machine [45,46,47,48]. This is also the purpose of the current paper.

In the case of difficulties in separating diagnostic (useful) information from vibration signals and the inability to formalize the mathematical algorithm for signal processing, the support of neural networks is used more and more frequently [49].

The components on the axes of a Cartesian coordinate system of accelerations were registered. It was assumed that the method must be simple and easily applicable in mobile devices. Thus, the analysis was conducted on the waveform of the vibration. The scope of the application included the identification of movement and dynamics, and evaluation of the technical state of the rail track. The two-stage analysis approach is proposed. Statistics and resultant vector methods are presented.

The main advantages of the presented study are described below:-A relatively simple way of collecting data;-Simple data structure for recording;-A low-cost measuring device can be used;-The measurement method is universal;-The assembly of the measuring device can be performed in a way that minimizes the impact of interferences, e.g., to the axis of the rim;

The main disadvantages of the presented study are described below:-The analysis requires the processing of large data sets,-The method does not identify the causes of the condition of the subgrade and rails;-At the moment, there are no studies in the field that could be used for verification.

## 2. Research Methods

As a continuation of the research conducted so far, a simple method was used to measure the vibrations of the driver’s cab presented in [2]. The research included two types of rail vehicles: a diesel locomotive (SM42) and a suburban multiple-unit (EN 57 SMUS) cabin, the latter of which performs shunting or another kind of activity. The measurements used a maintenance-free mobile application developed by the authors, installed on a mobile device located in the driver’s cab. The recording device was equipped with the standard functions of a MEMS (Micro Electro–Mechanical System) accelerometer and a GPS (Global Positioning System) receiver.

The research was carried out under two different conditions of the railway infrastructure and with two different train vehicles. Additionally, two different types of driving were investigated—one in the marshaling yard with the SM42 locomotive, and the second during access to the platform with the EN57 suburban multiple-unit set. These assumptions allowed us to extend the analysis and conclusions. The site plans of the testing workshops are depicted in Figure 1.

Figure 1A is a site plan of the Chabówka (Poland) station, i.e., the area where the studies in question were carried out, case 1 (49.595109° N, 19.926443° E). Figure 1B is a site plan of the Katowice Ligota (Poland) station, i.e., the area where the studies in question were carried out, case 2 (50.207551° N, 18.955843° E). In case 1, pilot studies were carried out using the diesel locomotive (Figure 2A,B). In case 2, pilot studies were performed using the EN57 suburban multiple-unit set (Figure 2C,D).

During the investigation, the accelerations of the driver’s cab vibrations were measured using proprietary software and a mobile device. The choice of the location of the measurement system resulted from accessibility to the cabin space, which, in the case of rail transport and difficult accessibility, as well as a very large variety of wagons, is very important. The proposed measurement method was simple and very practical, which in the future, may determine the applicability of this method in large-scale holistic research. The research scale effect is extremely important in the context of the purpose of the research, i.e., applicability of vibration signals for train and track real-time condition monitoring. In addition, this method allows for a continuous mode, which additionally enables an even more reliable correlation of the condition of the vibration measures with the condition of the rail infrastructure. Assuming the possibility of synchronizing the measurement system with SMS (Short Message Service) or GPRS (General Packet Radio Service) information channels, this method can be used in real-time technical condition monitoring systems.

The method assumed application in smart phones. Therefore, ADXL335 was used for vibration measurements due to its application on mobile devices. The ADXL335 is a small, thin, low-power, complete 3-axis accelerometer with signal-conditioned voltage outputs. The product measures acceleration with a minimum full-scale range of ±3 g. It can measure the static acceleration of gravity in tilt-sensing applications, as well as dynamic acceleration resulting from motion, shock, or vibration. As shown in Figure 3, ADXL335 measures the linear acceleration in three axes (Figure 3). The locomotive motion accelerations are designated as a_y_, lateral accelerations are designated as a_x_, and vertical accelerations as a_z_. At the same time, using the GPS receiver installed on the device, one reads the physical location of the locomotive within the range of vehicle positions (latitude and longitude). The test application was developed for the Android platform, Windows Phone systems, and iOS systems.

The mobile device consists of an accelerometer system used to record changes that occur in the linear accelerations of the driver’s cab within an assumed 10-millisecond interval (the interval can vary within a wide range of values). Furthermore, using the GPS system installed on the same device, the vibration characteristics can be correlated with the physical position of the railway vehicle performing the activity within the station track system (Figure 4). The locomotive trace recorded in the course of the studies is illustrated in Figure 4. As a result of follow-up analyses (not discussed in this article), the SM42 locomotive and EN 57 trace measurement allow evaluation of the vibration impact on the driver depending on the shunting or other types of operation performed.

## 3. Method of Analysis

It was assumed that, apart from driving dynamics, the condition of the railroad bed has the greatest impact on the recorded vibration accelerations. During vibration measurement, the signals and the current GPS position were recorded synchronously. The results obtained during the tests were compared with the traction characteristics of the tested vehicles, that is, the diesel locomotive and EN 57.

According to assumptions, the calculations were conducted based on the waveform of vibration. The purposes of the calculation were the identification of the movement and the dynamics and evaluation of the technical state of the rail track. The two-stage analysis approach is proposed. The statistics and resultant vector methods are presented.

Figure 4A illustrates the locomotive positions of the trace during its shunting activity at the station. The trace of the locomotive performing said operations has been marked in blue and red (pink) colours (the division is only due to technical and not subject-related reasons). In Figure 4, there are certain inaccuracies in the GPS signal measurements, namely the deviations between the locomotive positions and those of the infrastructure of the elements of the track (ranging up to several meters). However, it is not an obstacle to assigning the position of the locomotive to major infrastructure elements, as the vehicle was performing the shunting activity. In dubious cases, one may manually correct the relevant positions. Figure 4B illustrates the trace of the EN57 train positions during its activity at Katowice Ligota station. In this case, it is a typical local train (not shunting activity).

Both cases studied were experimental areas for the purpose of identifying the vibration properties of train vehicle. The first case included shunting processes in a small area with long-term duration. This type of activity generates numerous changes in driving direction with much point switching at a relative low speed. The second case referred to regular train driving with different speeds on the track, with more diverse railway infrastructure but fewer points that changed. For shunting activities, in addition to typical vibration generated by wheel–rail track contact and bogie suspension, vibration can be observed as a result of shocks from the coupling of vehicles. These phenomena are not observed during regular train driving (case 2).

The authors provide a comparison of linear accelerations in three axes recorded while running on different track sections, and different vehicles at the different stations, in the course of normal and shunting activity. All the data represent ca. 1.5 h of measurements. With this in mind, the figure below (Figure 5) shows the sample data for a 10 s research interval. Figure 5 illustrates the linear acceleration for three axes of the EN57 SMUS train positions during its activity at the Katowice Ligota station. In this case, it is a local train in typical course (not shunting activity). Figure 5 illustrates that the EN57 SMUS motion accelerations are designated as *a_y_* (green), the lateral accelerations as *a_x_* (orange), and the vertical accelerations as *a_z_* (blue). By analyzing the vibration, the current movement of the railway vehicle can be determined at selected points on the track. Thus, the following dynamical properties are observed: acceleration, deceleration, uniform speed, and left and right swing.

Figure 6 illustrates the linear acceleration for three axes of the locomotive positions during its activity at the Chabówka station. In this case, it is a shunting locomotive in a typical shunting course. Figure 6 illustrates that motion accelerations are designated as *a_y_* (green color), lateral accelerations as *a_x_* (orange color), and vertical accelerations as *a_z_* (blue color).

Based on the comparison of Figure 5 and Figure 6—linear accelerations recorded while running on different sections of the track at the station in the course of the activity of the railway vehicle—we can generalize some conclusions. Figure 5 and Figure 6 provide a collation of linear accelerations recorded in a smaller fragment of the track (with and without switches). This is only a slice and, at the same time, an example of how these characteristics look. These graphs show how vibrations from different elements of the track infrastructure are reflected in the data recorded by the measuring devices. Regarding the journey to the railway siding, one can clearly notice that the characteristic *a_x_* increases throughout the measurement range.

Figure 7 presents a visualization of vehicle positions for the characteristics drawn in Figure 6 in the geocentric reference system. Each section of the track can be remarked on in this manner. This problem boils down to the manipulation and statistical analysis of large data sets (BIG DATA).

The deviations from the zero value of lateral accelerations *a_x_* indicate the track roughness or geometrical nonlinearity of the rail tracks. An overlap of these irregularities in vertical and horizontal axes forced an observation of the complex characteristics of the vibration. Hence, the resultant vectors of acceleration and angle value are calculated. Each sample of recorded acceleration for which the standard deviation is over a defined limit has to be analyzed.

Figure 8 is a set of graphs that illustrates, in particular, the vibrations that occur on three axes of the rail vehicle, as recorded in the course of the tests. Figure 8A,C,E illustrate the vibrations occurring in the three axes of the EN57 driving at an average normal course speed. Figure 8B,D,F illustrate the vibrations occurring in the three axes of the locomotive that drives at an average shunting speed.

These figures show the relationships between linear acceleration and infrastructure. Additionally, the differences in speed and type of course (normal and shunting) are significant. Changes in the characteristics of the suspension have not been studied, and they are permanent over a short period. Figure 8 shows a comparison of vibration acceleration signals for two different driving speeds and two different sections of the railway line. The graphs show the vibrations on the longitudinal, transverse, and vertical axes. For the presentation of the results, vibration signals at junctions located in the immediate vicinity of the national road were selected (Figure 9A). Vibrations were observed on a railway bridge located over a small river (Figure 9B). These are typical places where higher vibrations can be expected.

As part of the preliminary analysis of the results presented in [2], the basic statistical measures were determined in successive time windows of the registered data sets. Table 1 presents examples of mean values and standard deviations as an illustration of the preliminary results analysis procedure. In the following lines of the table, the estimators were compiled for successive 1 s distributions of vibration acceleration values. The last line presents global measures determined for the entire recorded course of vibrations. Initial data analysis enables the comparison of instantaneous values with respect to each other and with respect to representative global values. With regard to synchronously recorded GPS signals, it is possible to track changes in the vibration value as a function of the current position of the train.

For more complex statistics analysis, the number of occurrences of acceleration values in the selected ranges was determined. Figure 10 shows three frequency observation graphs for linear accelerations recorded in all the axes. The range of variability of each linear acceleration was divided into 24 class ranges from −12 [m/s^2^] to +12 [m/s^2^]. This is a different approach to normal vibration signal processing. When a vibration spectrum is used, frequency distribution is more important and is often used. The vibration presents the distribution of the signal in the frequency domain, defined as the number of cycles per unit of time (1/T). Thus, it can be considered as a representation of the dynamical properties of the phenomenon. To identify abnormal dynamical states, in terms of acceleration value, the statistical method works well.

Figure 10A,C shows the most dominant class of the number of occurrences in the acceleration value. It refers to vibrations in the range of 8 ÷ 11 m/s^2^ on the *a_z_* and *a_x_* axes.

This graph may be referred to as a frequency observation of value vibrations in a defined class for a measuring section in which the locomotive conducted the shunting activity. This graph demonstrates the relatively homogeneous characteristics of acceleration in the direction of the locomotive travel (positive aspect). At the same time, one may notice typical abrupt changes to the accelerations occurring in the vertical axis, which may be a consequence of the technical condition of the track subgrade at the given station facility where the measurement was taken. Similar changes are typical of accelerations in the lateral axis of the rail vehicle.

The vibration characteristics in the context of the calculated statistical data sets depend on several factors, such as the track geometry (gradient), the suspension of the bogie, the ground and track condition, the type of rail vehicle, weather conditions, the remaining rolling stock (wagons) and much more. Equations (1) and (2), an orderly six:(1)$$a_{i}\left(t\right)=f\left(g_{1}\left(t\right),g_{2}\left(t\right),sp\left(t\right),sw\left(t\right),wa\left(t\right),ca\left(t\right),\left\{\varsigma \right\},\ldots \right),$$
(2)$$a_{i\left(t\right)}=<G,SP,SW,WA,CA,\Sigma >,$$
where:
*a_i_*(*t*)—acceleration in the i-th axis [m/s^2^];$G=\{g_{1}\left(t\right),g_{2}\left(t\right),\ldots ,g_{i}\left(t\right),\ldots g_{n}\left(t\right)\}$—various track geometry parameters, which also change over time;SP={sp(t)}—subgrade condition;SW={sw(t)}—suspension condition;WA={wa(t)}—atmospheric conditions;CA={ca(t)}—rolling stock load;Σ ={ς}—random factors, etc.

Measurements of train acceleration enable more advanced analysis. The authors decided to develop other useful metrics to describe the dynamical properties of the driving train. Due to the heavy construction of the train vehicle and the typical solution of the suspension system between wheelsets, bogies, and the cabin or wagon, the current position of the cabin must be considered as an important parameter. The position of the cabin (superelevation) is the result of the generation and interaction of the dynamic forces during the movement of the train. Thus, the authors propose another parameter for the evaluation of the current position of the cabin as the angle of resultant between the horizontal and vertical components of the current acceleration. It is calculated as the inverse tangent function of the current longitudinal and transverse; longitudinal and vertical; and transverse and vertical accelerations. As a result, we obtain the resultant angle and information on the relation between longitudinal, transverse, and vertical interactions. The distribution of the inverse tangent function is depicted in Figure 11.

The analysis of Figure 11 clearly shows the differences in the distribution of the inverse tangent functions for typical observation (normal trace) and some anomalies for the other trace (shunting or access to the platform). The graphs on the left side of Figure 11 represent the normal trace and the distribution is much more organized. The chart on the right side of Figure 11 is calculated for the shunting processes and is more chaotic. This can be interpreted as movement with different speeds and directions, or as shunting processes. This function can be useful for research on dynamic interactions in rail vehicles, because even small changes in interactions between the rail vehicle and the track are well recognized in the distribution of the inverse tangent function. It is much more visible than in linear acceleration observation.

The satisfactory results of the inverse tangent function provide further analysis. Hence, the authors decided to calculate the resultant vectors of accelerations. Two vectors were calculated; one was calculated in horizontal orientation as the sum of two longitudinal and transverse accelerations (Equation (3)). The second resultant vector was calculated as the sum of the first resultant vector and the vertical (Equation (4)). The calculation was as follows:(3)$$a_{x,y}=\sqrt{\vec{a_{x}^{2}}+\vec{a_{y}^{2}}}$$
(4)$$a_{z,xy}=\sqrt{\vec{a_{x,y}^{2}}+\vec{a_{z}^{2}}}$$
where:

a_i_—acceleration in the *i*-th axis [m/s^2^].

Figure 12 presents the result of the calculation. The red color function represents the resultant values of longitudinal and transverse accelerations (Equation (3)). It should be noted that for specifying the direction of the acceleration vectors, the assumption was accepted that the direction is consistent with the axis of movement of the vehicle. The green color function is the distribution of the values of the resultant vector of vertical and horizontal acceleration (Equation (4)).

Analyzing the distribution of the resultant vectors enabled the evaluation of the interaction of dynamical forces on the train. It was recognized that these functions are correlated with the technical condition of the rail infrastructure. The most significant impact was observed to be due to the longitudinal and transverse profile of the railway line. This allowed localization of route section on track where some dynamical anomalies were detected.

## 4. Track in Service Diagnostics

It should be noted that one can normalise the values of the vibration parameters where elements of the railway infrastructure are concerned. Under such normalization, it is possible to define an array of permissible vibrations for individual elements of known geometrical parameters and for known means of transport. Such an array may set the range of vibrations as permissible under typical track operating conditions: (5)$$A=[\Delta a_{x}\left(t,L\right);\Delta a_{y}\left(t,L\right);\Delta a_{z}\left(t,L\right)],\;$$
where:
$\Delta a_{i}^{}(t,L)$—range of correct (permissible) acceleration values along the *i*-th axis;t—age of track infrastructure elements;L={W,LO}—GPS geographical location of infrastructure elements (W-latitude, LO-longitude).

Any anomalies revealed in the course of the measurement in terms of the values envisaged as standard (Equation (5)) should be considered as a basis for a detailed analysis of the safety of rail traffic. Moreover, the act of recording vibrations that are stronger than permissible should lead to a reduction in the duration of locomotive driver shifts worked on track sections diagnosed as above.

During the experiments and analysis, the authors decided to extend the purpose of the methodology and developed a system for the infrastructure monitoring subsystem. Thus, the paper also discussed the possibilities of using the developed methods for the evaluation of railway infrastructure based on vertical and lateral vibration. The technical condition of the railway infrastructure—for example, the geometry of the track or the slab track—has an extremely important impact on the safety and efficiency of transport processes.

The measurement system and methodology allow control and registration of the current position of a train, with a ca. 1 s time interval (GPS module) [23,24]. The system also enables simultaneous registration 10 times for each 1 s acceleration of train (MEMs module). It allows one to locate the current space on the rail track occupied by the train. The precision of the identification of locomotive position depends on the current speed. For the analyzed case studies, a range was obtained of between 0.02 and 0.1 [m] (Table 2).

It should be noted that such a distance allows for a very precise identification of any problems in rail transport. If the sampling frequency of the recording data was higher, the track monitoring resolution would also be better. However, the authors did not investigate the influence of the sampling frequency on the possibility of determining the condition of the railway track or the subgrade. This is an interesting analysis of how to choose the frequency in order to effectively identify the state of the subtrack.

The concept of the infrastructure monitoring subsystem is based on analysis of the exceeded vibration level, which can be considered as a symptom of damage to the railway infrastructure. On the basis of the analysis of inverse tangent function and the resultant vector distribution, it was observed that the most significant impact occurs due to longitudinal and transverse accelerations. Thus, a statistical analysis of such a database was conducted. The focus of the analysis was on track-in-service diagnostics and detection of damage of rail tracks and elements of the rail infrastructure (e.g., shunting or remote-controlled points). It can be assumed that all anomalies in the presented characteristics generated by defects or wear processes will occur in the variance value.

Based on the fact that the developed method uses many characteristics simultaneously, heteroskedasticity can be analyzed in the proposed measuring system variables. Heteroscedasticity defines a collection of random variables with subpopulations that have different variabilities from others. Here, variability was quantified by variance. 

The second stage of the research included analysis of the track sections. For this purpose, the railway infrastructure of the testing workshop was described and grouped into four segments: A, B, C, and D. The selections were the type of infrastructure (e.g., the number of points or switches) and cross-slope terrain. The research was carried out at the Katowice Ligota station. The site plan of the test workshop with defined segments is shown in Figure 13.

For each segment, the mean value (Avg), variance (Var), mode (Mode), and median (Med) were calculated separately for the three directions of accelerations. The results of the calculations are depicted in Table 3.

Based on the statistical characteristics of a data set of acceleration in the defined segments, significant differences were found in the values of statistical estimators, depending on the direction of vibrations and the segment (track section) of the analysis. The tests were carried out in the area where the contact track was installed. The variability in the Y axis is of little interest from a research point of view, due to the fact that any such tests based on this parameter should use and coordinate data on the operating speed of the rail vehicle on the route. High values of variability are shown for accelerations in the *x*-axis, especially on sections B and D; however, in turn, such analyses require knowledge of the track profile and correlation of these data with each other.

The deviation of the basic statistical values for each segment allows one to assume that it is the result of differences in the railway infrastructure. The higher value of variance in segments B and D may be the result of the large number of points and curves of the tracks in the segments.

Table 3 includes characteristics of large segments; however, according to the precision of the identification of locomotive position (Table 2), it is even possible to collect the same data set for each 0.02 m section of track in service (for the speed of 10 km/h). Thus, this approach can be considered fundamental in performing diagnostics of track in service.

## 5. Discussion and Conclusions

This paper presents a complex collection of characteristics based on linear accelerations of a rail vehicle in operation. Registered signals are collected as large data sets. The measurement system includes a GPS module. Therefore, acceleration signals are correlated with the physical position of the rail vehicle and can be located on a geographical map and in the railway network (NMEA 183 protocol). 

For the authors, the purpose of this article was to describe the concept of an extremely simple method that enables holistic studies of the process of vibration interactions on the rail vehicle. Having satisfied specific requirements, one may conduct such a study in real time. Consequently, performing such a distributed type of study allows for statistically reliable conclusions to be formulated as regards the impact of the vibrations in question on the health condition of rail vehicle operators, and on others. The scheme of the complex research method is shown in Figure 14.

It was assumed that the method has to be easily applicable in mobile devices (e.g., smartphone, tablet). Thus, the analysis and calculation were based on the waveform of vibration. The application performs identification of train movement and dynamics, and an evaluation of rail track’s technical state. The mobile device features an accelerometer system and a GPS system.

By analyzing the vibration, the current movement of the railway vehicle can be determined at selected points on the track. Thus, such dynamical properties are observed: acceleration, deceleration, uniform speed, and left and right swing. The deviations from the zero value of lateral accelerations *a_x_* indicate the track roughness or geometrical nonlinearity of the rail tracks. An overlap of these irregularities in vertical and horizontal axes forces observation of the complex characteristics of vibration. Hence the resultant vectors of acceleration and angle value are calculated.

The method provides a sample statistical analysis of the measurement data obtained on the accelerometer. The data were collated with reference to mean values and standard deviation for linear accelerations recorded in individual axes. For the more complex statistical analysis, the number of occurrences in acceleration value class was determined. The authors decided to develop other useful metrics to describe the dynamical properties of the driving train. One of the proposed parameters for evaluating the current position of the cabin is the angle of the resulting horizontal and vertical accelerations. This was calculated as the inverse tangent function of current longitudinal and transverse; longitudinal and vertical; and transverse and vertical accelerations. As a result, we obtain the resultant angle and information on the relationship between longitudinal, transverse, and vertical interactions. In addition, the resultant vectors of accelerations were calculated—one in horizontal orientation as the sum of two longitudinal and transverse accelerations, and the second calculated as the sum of the first resultant vector and the vertical.

The last application of the method was performed as a track-in-service diagnostic. Based on general statistics, the values of triaxial accelerations calculated for the defined segments were recognized, as was the correlation with the type of rail infrastructure. Thus, this approach can be considered fundamental in performing diagnostics of track in service.

The results of these studies can also be used in the field of testing wheel–rail contact [50,51,52]. However, this type of test requires a slightly different position of the measuring instruments in the rail vehicle.

The results of these studies can also be used in the field of control of transport processes, considering that the vibrations of the vehicle–rail system allows one to determine the quality of the transport process in relation to:-The quality of the railroad with regard to performance and possible modernization, as well as renovation,-which in effect determines the quality of transport, especially in relation to goods sensitive to damage;-The condition of the means of transport;-The proper selection of the operation of the rolling stock for the conditions on the tracks-as well as for the transported cargo, etc.

The paper is the first presentation of the possibilities of the developed research method. It also presents examples of the dynamical characteristics of the rail vehicle and possibilities for analyzing the results. Additionally, new signal processing and statistical analysis tools have to be developed for the purpose of extracting more useful information.

## Figures and Tables

**Figure 1 sensors-22-02368-f001:**
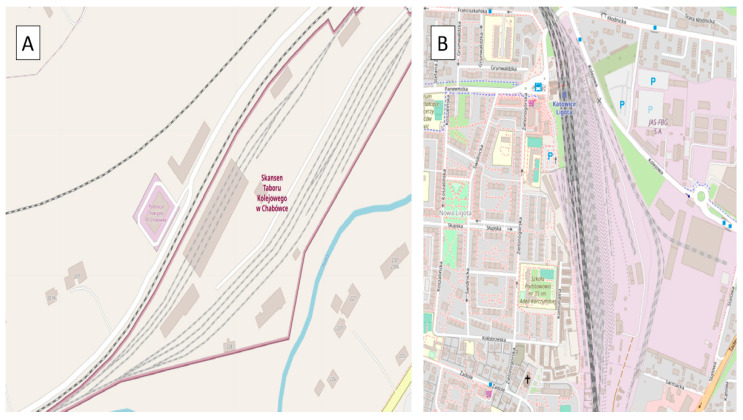
Site plans of the research stations where the measurement was conducted: (**A**) Chabówka, Poland; (**B**) Katowice Ligota, Poland. Source: OSM [50,51].

**Figure 2 sensors-22-02368-f002:**
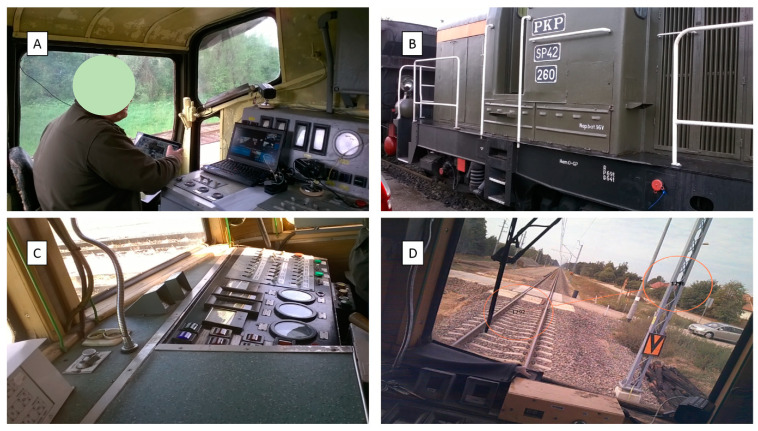
EN57 suburban multiple-unit set and SM42 locomotive used in the studies: (**A**) SM42 interior; (**B**) SM42 locomotive; (**C**) EN57 driver’s desk; (**D**) view from EN 57. Source: authors’ photographs.

**Figure 3 sensors-22-02368-f003:**
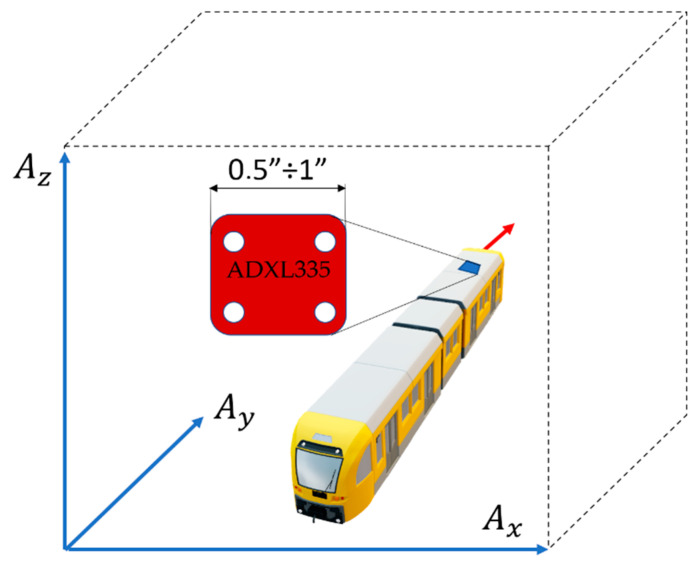
Small-size ADXL335 accelerometer with axes of acceleration sensitivity.

**Figure 4 sensors-22-02368-f004:**
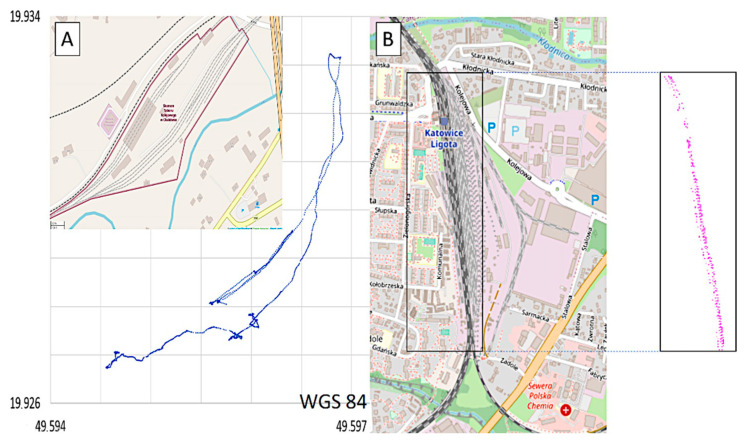
(**A**) /Trace of the locomotive performing shunting activity at the Chabówka station (**B**) /trace of EN57 suburban multiple-unit set train at the Katowice Ligota station. Source: OSM/JSOM [50,51,52].

**Figure 5 sensors-22-02368-f005:**
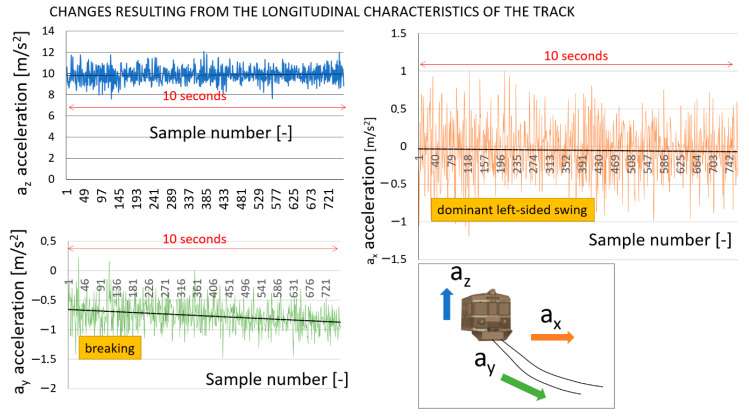
Visualization of the example data from the output file containing linear accelerations. EN 57 sample run in a straight section of the station track. Source: authors’ own materials.

**Figure 6 sensors-22-02368-f006:**
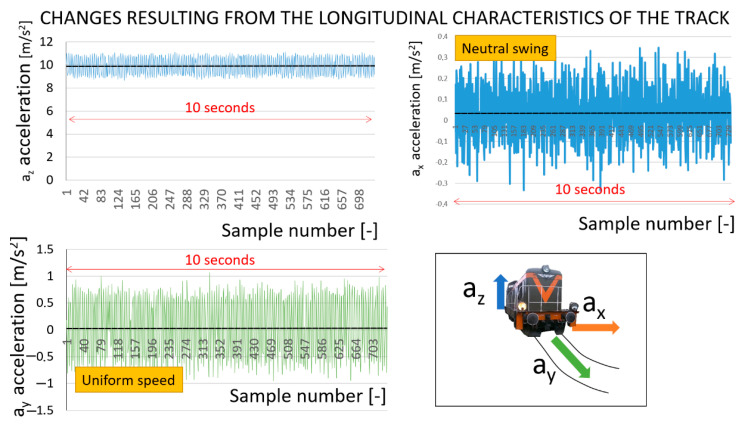
Visualization of example data from the output file containing linear accelerations. Locomotive sample shunting run in a track section featuring switches. Source: authors’ own materials.

**Figure 7 sensors-22-02368-f007:**
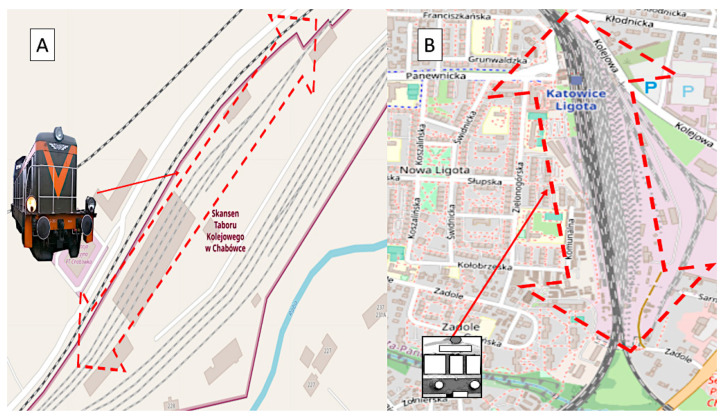
Geovisualization of example data in case 1 and case 2: (**A**) SM42; (**B**) EN 57. Source: authors’ own materials.

**Figure 8 sensors-22-02368-f008:**
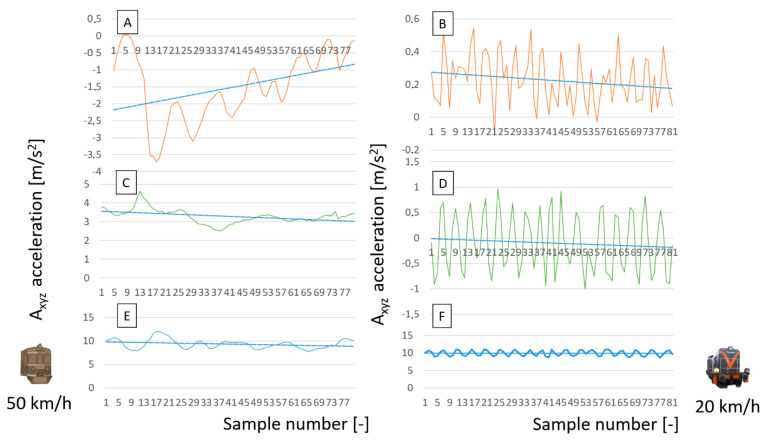
Selected examples of various vibrations experienced at various speeds and under different tasks: (**A**,**C**,**E**) Katowice Ligota station; (**B**,**D**,**F**) Chabówka station. Source: authors’ own materials.

**Figure 9 sensors-22-02368-f009:**
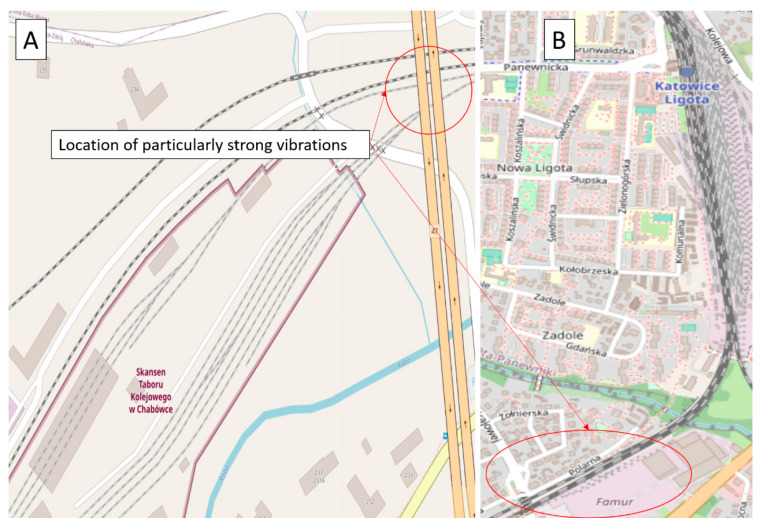
Location of particularly strong vibrations observed in the course of: (**A**) shunting activity at the Chabówka station; (**B**) normal course in Katowice Ligota. Source: authors’ own materials.

**Figure 10 sensors-22-02368-f010:**
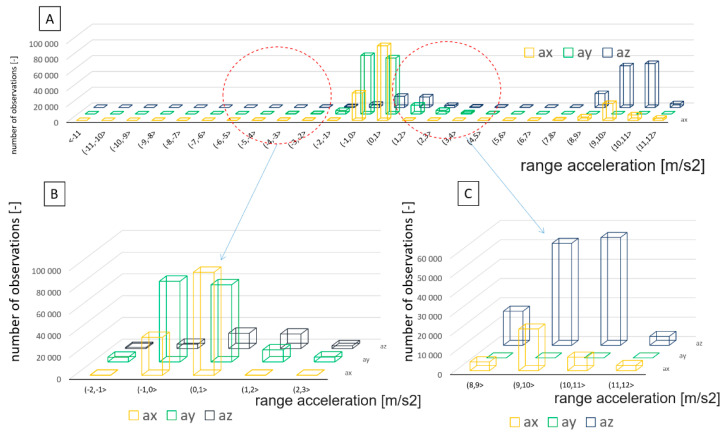
Frequency observation of linear accelerations values: (**A**) /overall distribution (**B**) /distribution at low accelerations (**C**) /distribution at high acceleration Source: authors’ own materials.

**Figure 11 sensors-22-02368-f011:**
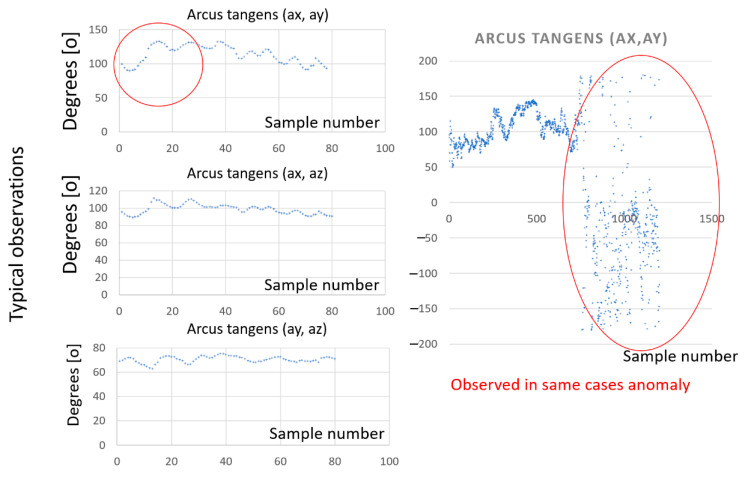
Distribution of inverse tangent function of accelerations: Source: authors’ own materials.

**Figure 12 sensors-22-02368-f012:**
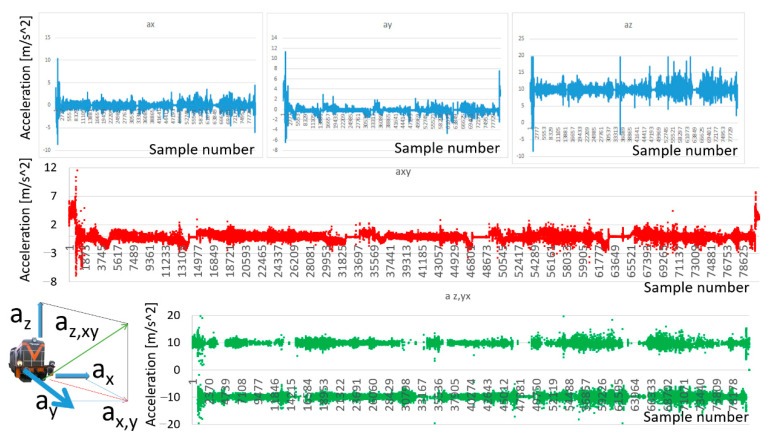
The resultant vectors of accelerations: Source: authors’ own materials.

**Figure 13 sensors-22-02368-f013:**
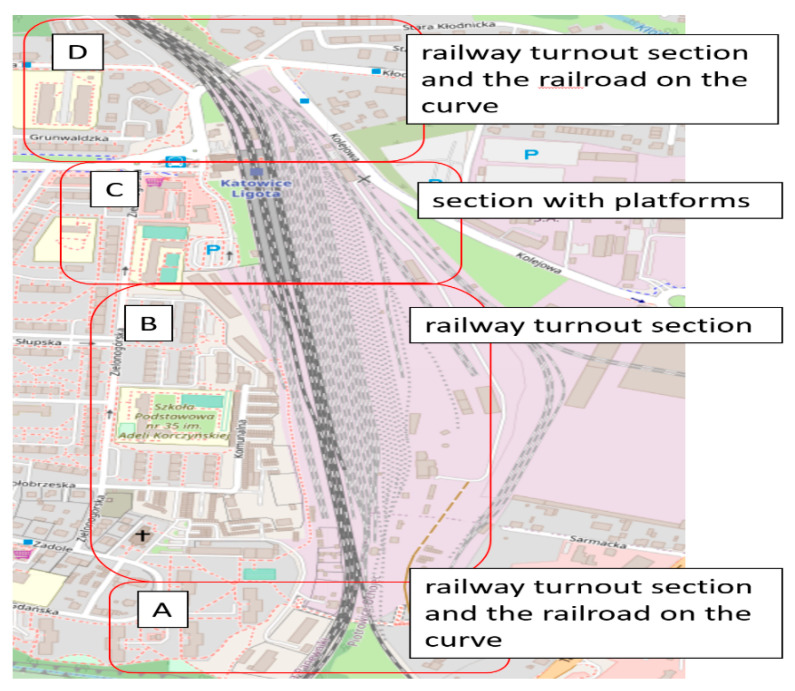
Site plan of testing workshop with defined segments, A, B, C, and D are selected groups of railway infrastructure of the testing workshop.

**Figure 14 sensors-22-02368-f014:**
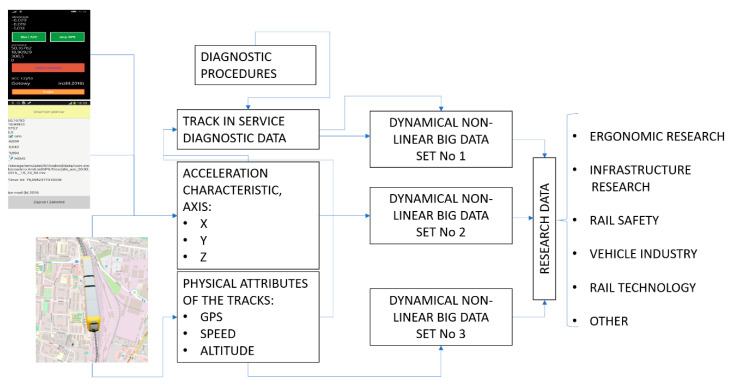
Research methodology scheme: Source: authors’ own materials.

**Table 1 sensors-22-02368-t001:** Basic statistical measures of vibration waveforms in successive time intervals. Source: adapted from [2].

Time Range [s]	Mean [m/s^2^]			Std [-]		
	*a_x_*	*a_y_*	*a_z_*	*a_x_*	*a_y_*	*a_z_*
**<0 ÷ 1)**	0.074	−0.003	10.056	0.134	0.571	0.762
**<1 ÷ 2)**	0.062	0.014	10.302	0.141	0.582	0.750
**…**	…	…	…	…	…	…
**…**	…	…	…	…	…	…
**…**	…	…	…	…	…	…
**<n – 1 ÷ n>**	0.520	2.359	9.747	0.284	0.140	0.785
** *total <0 ÷ n)* **	*2.160*	*0.139*	*7.861*	*4.061*	*1.036*	*3.918*

**Table 2 sensors-22-02368-t002:** The theoretical distance between measuring points of linear acceleration. Source: authors’ own elaboration.

km/h	m/s	a/10 ms [-]	ca. Δs [m]
**10**	2.7778	0.0278	0.02
**20**	5.5556	0.0556	0.05
**30**	8.3333	0.0833	0.08
**40**	11.1111	0.1111	0.11

**Table 3 sensors-22-02368-t003:** Statistical characteristics of a data set of acceleration in the defined segments. Source: authors’ own elaboration.

Value	Segment A			Segment B			Segment C			Segment D		
*x/y/z*	*a_x_*	*a_y_*	*a_z_*	*a_x_*	*a_y_*	*a_z_*	*a_x_*	*a_y_*	*a_z_*	*a_x_*	*a_y_*	*a_z_*
**Avg**	–0.32	–0.13	9.97	–0.31	–0.10	9.92	–0.28	–0.19	9.92	–0.10	–0.001	9.90
**Var**	0.13	0.006	0.04	0.35	0.03	0.20	0.09	0.15	0.07	0.45	0.06	0.18
**Mode**	–0.17	–0.13	9.99	–0.21	–0.17	9.99	–0.29	–0.11	9.92	0.00	–0.15	9.98
**Med**	–0.30	–0.13	9.91	–0.29	–0.10	9.91	–0.30	–0.11	9.91	–0.01	–0.06	9.88

## Data Availability

The data presented in this study are available on request from the corresponding author. The data are not publicly available due to company policy.

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
