# Peer review of "Research on the Applicability of Vibration Signals for Real-Time Train and Track Condition Monitoring [Author-notes fn1-sensors-22-02368]"

_sensors, 2022, doi:10.3390/s22062368_

Round 1

Reviewer 1 Report

The paper addresses the analysis of the possibilities of applicability of vibration signals to achieve real-time control of the train and runway.
For the research part, two types of railway vehicles were considered: a diesel locomotive (SM42) and a multiple suburban unit (EN 57 136SMUS), a cab, which performs maneuvering activities. for the experimental part, two cases were considered: one for short-distance maneuvering and speed variations and the second case for the regular driving of trains with different speeds on the rail, with a more diverse railway infrastructure. For maneuvering activities, in addition to the typical vibrations generated by rail contact and bogie suspension, vibrations can be observed as a result of vehicle coupling shocks. These phenomena are not observed during regular train operation The paper presents a series of characteristics based on the linear accelerations of the railway vehicle. Recorded signals are collected as large data sets. The measuring system includes a GPS module. Thus, the acceleration signals are correlated with the physical position of the railway vehicle and can be located on a geographical map and in the railway network (NMEA protocol 183).

Author Response

Responses:

Reviewer #1

Thank you very much for the review. The English language and style were improved.

Reviewer 2 Report

 I would suggest that the paper should be accepted with minor revision due to some of the corrections I pointed out in the attached file and in order to raise the standard of this paper. The English need to be polished, punctuation mark needs to be administered after each equation. Finally, I will be available for further revision of this
paper.

Author Response

Responses to the Reviewers

Thank you very much for your very insightful reviews and very valuable suggestions that allowed us to improve our article and will probably influence our future research.

Here are the contents of the review reports and responses:

Reviewer #2

Thank you very much for the review. All suggestions have been taken into account. The manuscript have been revised. Detailed responses and changes are presented below.

#1

The English need to be polished, punctuation mark needs to be administered after each equation.

The English language and style were improved. Punctuation mark were administered after each equation.

#2

Introduction

The opening paragraph of the paper will provide paper readers with their initial impressions about the logic of the paper argument, writing style, the overall quality /of research, and, ultimately, the validity of your findings and conclusions. Here, the introduction is required to improve by acknowledging some previous works. Also, in my opinion, the authors should highlighting the contributions of the cited papers. I suggest to improve the introduction section by adding some important works: https://doi.org/10.1186/s13662-021-03367-z

Introduction have been improved by acknowledging some previous works. Paragraphs marked in blue have been added.

Introduction

The current development, applications, and implications in the field of transportation are the main focus of transport scientists and engineers. The possibilities and impacts of emerging technologies, such as vibroacoustics, on transportation system performance, in terms of level safety and reliability, are widely discussed in many papers. Vibration phenomena in transport engineering can be considered as unwanted effects, mostly in terms of human exposure and as the source of degradation of infrastructure and suprastructure of transport system. But it should be considered as a source of many information on the technical condition of means of transport and transport network. Vibrations, very often of a non-linear nature, generated during the operation of machines and devices are transferred via hands or other parts of the body, including the feet and the lumbar region, in a sitting position to the human body. During long-term human exposure to vibrations, they may adversely affect your health [1-4]. The negative effects (pathologies) of long-term exposure to mechanical vibrations may be permanent or even irreversible. To a large extent, this risk depends on the frequency of vibrations, the exposure time, and the contact surface in the human-device system [3-6]. Conditions related to the negative influence of acceleration are generally called kinetosis [7,8]. The term vibration stress is also often encountered. In this way, the human body responds naturally to selected movement stimulants [11]. Simultaneously, vibrations transfer much information. The wide range of frequency capacity and non-destructive methods of vibration measurements are commonly used in machine monitoring systems [8-18]. Thus, vibroacoustic methods should be considered as support for monitoring systems in the field of transportation, due to their usefulness for the evaluation of human exposure and technical state monitoring of the transport system. The article presents implications of methodologies developed for the rail transport system for operation, control, maintenance, and rehabilitation services [25-27].

The paper [19] describes research on vibrations in the cab of the train driver during shunting activity. The authors extended the research and analysis for the purpose of applicability of the vibration signals of train driver's cab for train and track control. For this reason, in this article, the authors have addressed and discussed selected aspects of analysis of the vibration impact on a shunting locomotive and an EN57 suburban multiple-unit set driver.

One group of applications of vibration research refers to the evaluation of human exposure to vibration. It was the main focus of the paper [19]. In publications concerning this subject, problems of the vibration impact on people are mainly analysed with regard to manually operated machines and devices due to the sale of the phenomenon. In the cases when the vibration impact affects the entire body, it may primarily cause organ dysfunctions of the organs of the abdominal cavity (especially the stomach), whereas it also affects the abdomen in women. Other effects observed in practice on people operating in vibrating system include fatigue, often prolonged, headaches, sleep dysfunctions, and trembling limb symptom, typical of those operating vibrating equipment. However, in the case discussed, that is, the whole-body impact of vibrations, this problem may not only concern hands and arms [6-9,11,12]. Prolonged exposure of the human body to vibrations may trigger the spread of disorders to other internal organs. The fact that the effects of vibrations can cause diseases of the cardiovascular, respiratory, digestive, and musculoskeletal systems has been confirmed [6-9,11,12]. Vibrations that affect the entire body of a machine operator may also lead to changes in pulse, as well as in the vascular and urinary system. In addition to the direct hazard to the health and life of a worker exposed to whole-body vibrations, one can also observe a reduction in the immunity of the latter. Therefore, from the employer’s perspective, which involves maintaining continuity and operating safety, hazards connected with vibrations occurring in railway vehicles (or, more broadly, in transport) can be divided into short-term, which can involve hazard to traffic safety in certain circumstances, and long-term, causing deterioration of operating efficiency and loss of highly qualified workers for prolonged periods due to sick leave [6].

Vibrations are also used in systems for monitoring and diagnosing the condition of tracks and bogies of rail vehicles. Vibration of a train track under the train in motion provides important information about the train and the track. Traditionally, track monitoring systems are used for wheel impact load detection, continuous monitoring of wheel profile, bogie performance, etc. [28]. The paper [29] presents the concept of a simple sensor-based system for train condition monitoring. This system uses an ADXL335. This is a small, thin, low-power, complete 3-axis accelerometer with signal-conditioned voltage output. The fundamental time-domain and frequency-domain based methods have been used to determine the condition of a train using measurements of a railway track vibrations under train in motion. This article [29] confirms the legitimacy of searching for new, low-cost solutions that can be used in the diagnosis of tracks and rail vehicles. The authors conducted many studies using rail vibration measurements in the aspect of identifying passing trains [30] and vibration responses of rail infrastructure devices or passing cars [31], as well as identifying rail damage [32].

Another approach is to record vibrations in moving locomotives or wagons. The paper [33] presents the current state of the art and reviews recent on-board condition monitoring sensors and techniques and potential application for integrating condition monitoring systems into freight wagons. The paper [34] presents a simple and effective fault detection framework for the polygonal wear of the railway wheel. It is based on easily acquired axle box vibration acceleration data to detect both the level and the order parameters of the polygon wear of railway wheel. The paper [35] presents numerical experiments on the method that provides a useful reference source for the detection of polygonization for high-speed trains, as well as the judgement of the validity of dynamic response data. Another approach to vibration-based damage identification and condition monitoring of metro trains by the system installed in buildings has been presented in [36]. The vibration is triggered by the signal transmitted from the measuring point located in the tunnel, which triggers the registration with a photocell when passing a subway train. Therefore, all vibration methods used in the research on rail vehicles and tracks are based on dynamic dependencies and vibration signals generated as a result of them [37,38].

When considering the use of vibration signals for diagnostic purposes, acoustic measurements should also be considered as an additional source of information [39]. In this case, vibroacoustic methods can be applied. The paper [40] presents the application of distributed acoustic sensor technology to monitor the running condition of trains. The new method of monitoring train running conditions using distributed acoustic sensor (DAS) technology is based on the fact that the vibration of the environment around the optical fiber will change when the train passes [41]. This method is very effective, but requires a lot of money and cannot always be used. Therefore, the authors decided to carry out measurement experiments under the assumption of minimal measurement and equipment costs.

For railway traffic is enormously influenced by the variation in contact stress caused by wheel rail contact profile changes [42]. The paper [43] presents a study on the influence of the topology of the wheel and rail profile of standard rail and standard wheel profile. The condition of the wheel sets also significantly influences the vibrations generated by the rail vehicle. The paper [44] especially concerns abrasive wear of the wheel rims and damage of the spalling type on the wheel thread in wheel sets. Therefore, it should be emphasized how important in the research of rail vehicle vibrations is the phenomenon of wheel rail contact, which is the subject of many scientific publications.

Other types of vibration research focus on selection of useful information about the current condition of the machine [42-45]. It was also the purpose of the paper presented.

In the case of difficulties in separating diagnostic (useful) information from vibration signals and the inability to formalize the mathematical algorithm for signal processing, the support of neural networks is used more and more frequently [49].

The components on the axes of a Cartesian coordinate system of the accelerations have been registered. It was assumed that the method must be simple and easily applicable in mobile devices. Thus, the analysis was conducted on waveform of vibration. The scope of the application included the identification of the movement and the dynamics and evaluation of the technical state of the rail track. The two-stage analysis approach is proposed. Statistics and resultant vector methods are presented.

 Main advantages of the presented study are described below:

  • relatively simple way of collecting data,
  • simple data structure for recording,
  • a low-cost measuring device can be used,
  • the measurement method is universal,
  • the assembly of the measuring device can be performed in a way that minimizes the impact of interferences, g. to the axis of the rim,

Main disadvantages of the presented study are described below:

  • the analysis requires the processing of large data sets,
  • the method does not identify the causes of the condition of the subgrade and rails,
  • at the moment there are no studies in the field that could be used for verification.
  •  

Added new references:

  1. Sładkowski A., Bizoń K. Aspects of Rail Infrastructure Design. In: Marinov M. (eds) Sustainable Rail Transport. Lecture Notes in Mobility. Springer, Cham., 2018, https://doi.org/10.1007/978-3-319-58643-4_8
  2. Srivastava, J.P., Sarkar, P.K. & Ranjan, V. Contact Stress Analysis in Wheel–Rail by Hertzian Method and Finite Element Method.  Inst. Eng. India Ser. C95, 319–325 (2014). https://doi.org/10.1007/s40032-014-0145-x
  3. Guzowski S, Michnej M, Zając G. Tribological wear of wheel rims in rail vehicles in operating conditions. Tribologia. 2016, pp. 91-99.

  1. Rajchakit, G., Sriraman, R., Boonsatit, N. et al.Global exponential stability of Clifford-valued neural networks with time-varying delays and impulsive effects. Adv Differ Equ 2021, 208, https://doi.org/10.1186/s13662-021-03367-z
  2. Kisilowski, J.; Kowalik, R. Mechanical Wear Contact between the Wheel and Rail on a Turnout with Variable Stiffness. Energies202114, 7520. https://doi.org/10.3390/en14227520
  3. Trummer, G.; Lee, Z.S.; Lewis, R.; Six, K. Modelling of Frictional Conditions in the Wheel–Rail Interface Due to Application of Top-of-Rail Products. Lubricants20219, 100. https://doi.org/10.3390/lubricants9100100
  4. Turabimana, P.; Nkundineza, C. Development of an On-Board Measurement System for Railway Vehicle Wheel Flange Wear. Sensors202020, 303. https://doi.org/10.3390/s20010303

#3

Main Results

All results and presentations are well-defined. Answer the following queries:

  • How to fixup Eqn.[1]?
  • It is very important to know the advantages of present study? So, the introduction needs to improve by highlighting the main advantages?
  • I could not understand that how authors fix the formula given by Eqn. [4] ?

Eqn. [1] - clarified and extended the description, like follow:

The vibration characteristics in the context of the calculated statistical data sets depend on several factors, such as track geometry (gradient), suspension of bogie, ground and track condition, type of rail vehicle and even weather conditions, remaining rolling stock (wagons) and much more (Eqn. 1 and 2), an orderly six:

 , (1)

 , (2)

, where:

ai(t)

-

acceleration in the ith axis [m/s2],

-

various track geometry parameters, they also change over time

SP={

SW={

WA={wa

CA={ca

-

-

-

-

-

subgrade condition,

suspension condition,

atmospheric conditions,

rolling stock load

random factors etc.

The introduction section have been extended by highlighting the main advantages of the study. To be fair we have been considering also main disadvantages of presented study. Paragraphs marked in blue have been added.

Main advantages of the presented study are described below:

  • relatively simple way of collecting data,
  • simple data structure for recording,
  • a low-cost measuring device can be used,
  • the measurement method is universal,
  • the assembly of the measuring device can be performed in a way that minimizes the impact of interferences, g. to the axis of the rim,

Main disadvantages of the presented study are described below:

  • the analysis requires the processing of large data sets,
  • the method does not identify the causes of the condition of the subgrade and rails,
  • at the moment there are no studies in the field that could be used for verification.

The Eqn. (4) (after revision 5) represents normalization array of permissible vibrations for individual elements of known geometrical parameters of track infrastructure elements. Such an array may set the range of vibrations being permissible under typical track operating conditions. Any anomalies revealed in the course of the measurement in terms of the values envisaged as standard (Eqn. 5) should be considered as a basis for detailed analysis.

The formula given by Eqn. (4) - after renumbering (5) has been corrected.

Such an array may set the range of vibrations being permissible under typical track operating conditions:

] ,          (5)

, where:

-

range of correct (permissible) acceleration values along the i-th axis,

-

age of track infrastructure elements,

-

GPS geographical location of infrastructure elements (W - latitude, LO - longitude).

#4

References

I would recommend that the authors should check the correct way of citing the papers in their reference lists, or if possible they should check the website of the journals that published those papers instead of relying only on the information found on google scholar. In addition, the authors did not follow the reference style of this journal.

Thank you for paying attention. Reference style have been revised.

#5

My Recommendation

I would suggest that the paper should be accepted with minor revision due to some of the corrections I pointed above and in order to raise the standard of this paper. The English needs to be polished, punctuation mark needs to be administered after each equation. Finally, I will be available for further revision of this paper.

Thank you for such kind recommendation. All suggestions have been considered and the paper have been revised. This undoubtedly allowed us to raise the standard of this paper.

Reviewer 3 Report

Minor revision:

- extend the Introduction with current state on wheel-rail contact research paper.

- comment the results in Table 2.

- comment the result in Table 3.

- refer to the possibility of using the described method in the control of transport processes.

Author Response

Responses to the Reviewers

Thank you very much for your very insightful reviews and very valuable suggestions that allowed us to improve our article and will probably influence our future research.

Here are the contents of the review reports and responses:

Reviewer #3

Thank you very much for the review. All suggestions have been taken into account. The manuscript have been revised. Detailed responses and changes are presented below.

#1

Minor revision:

- extend the Introduction with current state on wheel-rail contact research paper.

Thank you for this suggestion. The Introduction and Discussion and Conclusions sections have been extended with current state on wheel-rail contact research paper. Paragraphs marked in blue have been added.

Introduction

For railway traffic is enormously influenced by the variation in contact stress caused by wheel rail contact profile changes [42]. The paper [43] presents a study on the influence of the topology of the wheel and rail profile of standard rail and standard wheel profile. The condition of the wheel sets also significantly influences the vibrations generated by the rail vehicle. The paper [44] especially concerns abrasive wear of the wheel rims and damage of the spalling type on the wheel thread in wheel sets. Therefore, it should be emphasized how important in the research of rail vehicle vibrations is the phenomenon of wheel rail contact, which is the subject of many scientific publications.

Discussion and Conclusions

The results of these studies can also be used in the field of testing wheel-rail contact [50-52]. However, this type of test requires a slightly different position of the measuring instruments in the rail vehicle.

The results of these studies can also be used in the field of control of transport processes. Considering the vibrations of the vehicle-rail system allows one to determine the quality of the transport process in relation to:

  • quality of the railroad, with regard to performance and possible modernization, as well as renovation,
  • which in effect determines the quality of transport, especially in relation to goods sensitive to damage,
  • condition of the means of transport,
  • proper selection of the operation of the rolling stock to the conditions on the tracks
  • also for the transported cargo, etc.

Added new references:

  1. Sładkowski A., Bizoń K. Aspects of Rail Infrastructure Design. In: Marinov M. (eds) Sustainable Rail Transport. Lecture Notes in Mobility. Springer, Cham., 2018, https://doi.org/10.1007/978-3-319-58643-4_8
  2. Srivastava, J.P., Sarkar, P.K. & Ranjan, V. Contact Stress Analysis in Wheel–Rail by Hertzian Method and Finite Element Method.  Inst. Eng. India Ser. C95, 319–325 (2014). https://doi.org/10.1007/s40032-014-0145-x
  3. Guzowski S, Michnej M, Zając G. Tribological wear of wheel rims in rail vehicles in operating conditions. Tribologia. 2016, pp. 91-99.

  1. Rajchakit, G., Sriraman, R., Boonsatit, N. et al.Global exponential stability of Clifford-valued neural networks with time-varying delays and impulsive effects. Adv Differ Equ 2021, 208, https://doi.org/10.1186/s13662-021-03367-z
  2. Kisilowski, J.; Kowalik, R. Mechanical Wear Contact between the Wheel and Rail on a Turnout with Variable Stiffness. Energies202114, 7520. https://doi.org/10.3390/en14227520
  3. Trummer, G.; Lee, Z.S.; Lewis, R.; Six, K. Modelling of Frictional Conditions in the Wheel–Rail Interface Due to Application of Top-of-Rail Products. Lubricants20219, 100. https://doi.org/10.3390/lubricants9100100
  4. Turabimana, P.; Nkundineza, C. Development of an On-Board Measurement System for Railway Vehicle Wheel Flange Wear. Sensors202020, 303. https://doi.org/10.3390/s20010303

#2

- comment the results in Table 2.

Thank you for this suggestion. Paragraph marked in blue have been added.

It should be noted that such a distance allows for a very precise identification of any problems in the rail transport. If the sampling frequency of the recording data was higher, the track monitoring resolution would also be better. However, the authors did not investigate the influence of the sampling frequency on the possibility of determining the condition of the railway track or the subgrade. This is an interesting analysis of how to choose the frequency in order to effectively identify the state of the subtrack.

#3

- comment the result in Table 3.

Thank you for this suggestion. Paragraph marked in blue have been added.

Based on the statistical characteristics of a data set of acceleration in the defined segments, significant differences were found in the values of statistical estimators depending on the direction of vibrations and the segment (track section) of the analysis. The tests were carried out in the area where contact track is installed. The variability in the Y axis is of little interest from a research point of view. Due to the fact that any such tests based on this parameter should use and coordinate data in the field of operating speed of the rail vehicle on the route. High values of variability is shown for accelerations in the x-axis, especially on the sections B and D, but such analyzes, in turn, require knowledge of the track profile and correlation of these data with each other.

#4

- refer to the possibility of using the described method in the control of transport processes.

Thank you for this suggestion. Paragraph marked in blue have been added.

The results of these studies can also be used in the field of control of transport processes. Considering the vibrations of the vehicle-rail system allows one to determine the quality of the transport process in relation to:

  • quality of the railroad, with regard to performance and possible modernization, as well as renovation,
  • which in effect determines the quality of transport, especially in relation to goods sensitive to damage,
  • condition of the means of transport,
  • proper selection of the operation of the rolling stock to the conditions on the tracks
  • also for the transported cargo, etc.